# *Raphanus sativus* Linne Protects Human Nucleus Pulposus Cells against H_2_O_2_-Induced Damage by Inhibiting TREM2

**DOI:** 10.3390/biology13080602

**Published:** 2024-08-09

**Authors:** Hyunseong Kim, Changhwan Yeo, Jin Young Hong, Wan-Jin Jeon, Hyun Kim, Junseon Lee, Yoon Jae Lee, Seung Ho Baek, In-Hyuk Ha

**Affiliations:** 1Jaseng Spine and Joint Research Institute, Jaseng Medical Foundation, Seoul 06110, Republic of Korea; biology@jaseng.org (H.K.); duelf2@jaseng.org (C.Y.); vrt23@jaseng.org (J.Y.H.); cool2305@jaseng.org (W.-J.J.); khyeon94@jaseng.org (H.K.); excikind@jaseng.org (J.L.); goodsmile@jaseng.org (Y.J.L.); 2College of Korean Medicine, Dongguk University, 32 Dongguk-ro, Ilsandong-gu, Goyang-si 10326, Republic of Korea; baekone99@gmail.com

**Keywords:** TREM2, nucleus pulposus, oxidative stress, *Raphanus sativus* Linne, degenerative disc

## Abstract

**Simple Summary:**

Intervertebral disc degeneration (IDD) occurs due to damage and loss of nucleus pulposus (NP) cells. This study investigated the protective effects of *Raphanus sativus* Linne (RSL), commonly known as radish, against oxidative stress caused by H_2_O_2_ in human NP cells and its ability to inhibit TREM2, a protein that induces apoptosis and degeneration in NP cells. RSL significantly improved cell survival by reducing the apoptosis markers cleaved caspase-3 and Bax and increasing Bcl2. It also enhanced mRNA levels of *ACAN* and *Col2a1*, important for NP cell function, and reduced levels of *ADAMTS-4, ADAMTS-5*, *MMP3*, and MMP13, which are involved in NP cell degeneration. Additionally, RSL downregulated TREM2 expression, which is elevated by H_2_O_2_ and causes disc degeneration. Overall, RSL extracts support human NP cells under oxidative stress and may help prevent IDD by regulating key degeneration pathways, particularly TREM2.

**Abstract:**

Intervertebral disc degeneration (IDD) progresses owing to damage and depletion of nucleus pulposus (NP) cells. Cytoprotection mitigates oxidative stress, nutrient deprivation, and mechanical stress, which lead to cell damage and necrosis. We aimed to examine the protective effect of *Raphanus sativus* Linne (RSL), common radish, against oxidative stress by H_2_O_2_ in human NP cells and whether the RSL extracts can inhibit triggering receptor expressed on myeloid cells 2 (TREM2), an inducer of apoptosis and degeneration in NP cells. We administered hydrogen peroxide (H_2_O_2_) to cultured human NP cells treated with RSL extracts. We used immunoblotting and quantitative PCR to investigate expression of the apoptosis-associated proteins in cultured cells. RSL significantly enhanced cell survival by suppressing the activation of cleaved caspase-3 and Bax. In contrast, RSL extract increased Bcl2 concentration to downregulate apoptosis. Additionally, RSL treatment notably enhanced the mRNA levels of *ACAN* and *Col2a1* while significantly reducing those of *ADAMTS-4*, *ADAMTS-5*, *MMP3*, and *MMP13*, key genes involved in NP degeneration. While H_2_O_2_ elevated TREM2 expression, causing disc degeneration, RSL downregulated TREM2 expression. Thus, our findings imply that RSL supports human NP cells under oxidative stress and regulates the pathways underlying disc degeneration, particularly TREM2, and that RSL extracts may potentially prevent IDD.

## 1. Introduction

Intervertebral disc degeneration (IDD) significantly impacts millions of people worldwide as it causes chronic back pain and diminishes quality of life [1,2,3]. Nucleus pulposus (NP) cells are vital for maintaining healthy and properly functioning intervertebral discs [4,5]. They hydrate the discs by producing mainly type II collagen and proteoglycans that retain water, enabling the discs to absorb and distribute the mechanical loads efficiently [6,7,8]. NP cells produce and regulate other extracellular matrix (ECM) components (including hyaluronic acid, fibronectin, and laminin) which are crucial for disc resilience during physical stress [9,10,11]. However, during disc degeneration, the viability of these cells and matrix production are compromised as a result of aging, mechanical overload, and oxidative stress [12,13]. Oxidative stress occurs when there is an imbalance between the production of reactive oxygen species (ROS) and the body’s ability to detoxify these harmful byproducts [14]. In NP cells, elevated ROS levels can trigger apoptosis, reducing the cell population necessary for maintaining disc integrity [5]. This apoptosis weakens the disc’s structure by decreasing the number of functional cells that produce essential extracellular matrix (ECM) components [15]. Mechanical pressure on intervertebral discs can result from factors such as heavy lifting, poor posture, and repetitive motions [16]. Repeated mechanical stress causes microtrauma to NP cells and the surrounding ECM, initiating a cascade of degenerative changes [17]. This mechanical damage accelerates the aging process of NP cells, further compromising their ability to maintain the ECM [18]. Together, oxidative stress and mechanical pressure contribute to the deterioration of intervertebral discs. This deterioration leads to conditions such as chronic back pain, which are characteristic of IDD. Therapeutic strategies to preserve NP cell function against a variety of stressors can help maintain the overall health and functions of the discs [19].

In this research, we focused on evaluating therapeutic strategies for inhibiting apoptosis and enhancing survival of the human NP cells by using cytoprotective agents that reduce oxidative stress and delay the degenerative process. We focused on the potential therapeutic benefits of herbal extracts, celebrated for antioxidative and anti-inflammatory effects and cytoprotective properties, which are studied for their potential to treat various degenerative diseases. For example, roots, seeds, flowers, sprouts, and leaves of *Raphanus sativus* Linne (RSL), commonly known as radish, have been extensively studied owing to the abovementioned beneficial effects. RSL extracts contain phytochemicals, such as flavonoids, phenolic compounds, and anthocyanins, which help scavenge free radicals and reduce oxidative stress [20]. Additionally, RSL exerts significant anti-inflammatory properties by inhibiting the NF-κB pathways, crucial for regulating inflammatory responses [21]. Notably, the cytoprotective properties of a specific fraction of the radish-leaf extract (F2) reportedly protected human fetal lung fibroblasts against oxidative damage induced by H_2_O_2_ [22]. Accumulated evidence showing that RSL reduces oxidative stress and inflammation supports its therapeutic use potentially for preserving properly functioning and healthy NP cells, crucial for preserving the structural integrity of intervertebral discs.

We explored the protective effects of the RSL seed extracts on human NP cells challenged by H_2_O_2_, aiming to demonstrate that RSL extracts can support cell survival and reduce cell death, ultimately safeguarding cells against oxidative stress. Specifically, we investigated whether the RSL extracts mitigate H_2_O_2_-induced oxidative damage in challenged NP cells by triggering receptor expressed on myeloid cells 2 (TREM2). TREM2 signaling is implicated in the occurrence of various neurodegenerative diseases [23,24,25]. TREM2, predominantly located on microglial surfaces, is integral to the immune system’s functionality [26]. Accordingly, TREM2 expression was reportedly higher in NP tissues from patients with IDD. Silencing TREM2 in human degenerative NP cells with siRNA reportedly decreased apoptosis, increased cell proliferation, and reduced secretion of inflammatory cytokines, including interleukin (IL)-6, tumor necrosis factor-alpha (TNF-α), and IL-1beta (IL-1β). These effects were mediated by upregulation of B-cell lymphoma (Bcl)-2 levels, downregulation of Bcl-2-associated X protein (Bax) levels, and reduced nuclear translocation of NF-κB p65. Conversely, TREM2 activation accelerated cellular degeneration. These effects were observed also in rat NP tissues with IDD, suggesting that inhibiting TREM2 represents a potential treatment for IDD [27]. Furthermore, TREM2 is involved in the cellular response to oxidative stress. For instance, in Alzheimer’s disease, TREM2 deficiency is associated with impaired microglial function, increased oxidative damage, and exacerbated neurodegeneration [28,29]. Another study indicates that TREM2 can inhibit the inflammatory response by reducing oxidative stress through activation of the PI3K/AKT pathway [30]. This underscores the protective role of TREM2 in mitigating oxidative stress and maintaining cellular and tissue integrity. 

Thus, we focused on examining the critical role of RSL extracts in regulating TREM2 and human NP cell functions, specifically assessing the effect of the extracts on cell viability or apoptosis through cleaved caspase-3, Bax, Bcl-2, and degeneration markers such as *ACAN*, *Col2A1*, *ADAMTS4*, *ADAMTS5*, *MMP3*, and *MMP13* under oxidative stress induced by H_2_O_2_. This study underscores the efficacy of RSL in promoting survival and inhibiting cell death in human NP cells under oxidative stress, thereby mitigating degeneration. It proposes that RSL could act as a cytoprotective agent, offering a potential strategy to prevent disc degeneration and bolster efforts to combat IDD.

## 2. Materials and Methods

### 2.1. Culturing of NP Cells

In this study, we acquired human NP cells from ScienCell Research Laboratories (Carlsbad, CA, USA). The NP cells were propagated using a dedicated Nucleus Pulposus Cell Medium, provided by the same company. This medium was enriched with fetal bovine serum at a concentration of 2% by volume, NP cell-specific supplements at 1% by volume, and an antibiotic cocktail of penicillin and streptomycin, also at 1% by volume. For our experiments, we utilized cells between the 3rd and 5th passages.

### 2.2. Preparation of RSL Extracts

RSL seeds, in their dried form, were purchased from Green M. P. Pharmaceutical Co., Ltd. (Sungnam, Republic of Korea). RSL seeds (30 g) were boiled in distilled water (300 mL) for 3 h at 105 °C in a heating mantle (Misung Scientific Co. Ltd., Gyeonggido, Republic of Korea). After boiling, they were filtered with a paper filter (Hyundai Micro, Seoul, Republic of Korea) and a vacuum pump (GAST, Benton Harbor, MI, USA) to remove the particulate matter. And then, RSL extract was frozen at −70 °C before being lyophilized in a freeze-dryer (Ilshin BioBase Co., Ltd., Gyeonggido, Republic of Korea) to dry the extract. The dry extract was normally stored at −20 °C unless used in experiments.

### 2.3. RSL or H_2_O_2_ Treatment

RSL stock (concentration of 20 mg/mL) was prepared in phosphate-buffered saline (PBS). Before any treatments, NP cells were seeded and given 1 day to attach and proliferate. After cells reached 60–70% confluence, they were treated with desired RSL concentrations, 400 μM H_2_O_2_ alone or pre-treated with RSL for 30 min followed by treatment with H_2_O_2_ for 24 h.

### 2.4. Cell Viability Assays

To assess the impact of RSL on cell viability, a Cell Counting Kit 8 (CCK8) assay (Dojindo, Kumamoto, Japan) was used. Initially, NP cells were seeded in 96-well plates filled with the culture medium. These were then exposed to a gradient of RSL concentrations from 1 to 400 μg/mL and incubated for 24 h. Following this period, each well received CCK8 solution mixed with 10% of the culture medium and was observed for 4 h using an Epoch microplate reader (BioTek, Winooski, VT, USA). The optical density was then determined at 450 nm and cell viability was quantified by comparing the mean percentage of surviving cells in the treated wells to those in control wells, using the following formula to calculate viability: $$Percentage\;of\;cell\;viability=(Mean\;OD\left(treatedc\;ells\right)÷Mean\;OD\left(control\;cells\right))\times 100$$

### 2.5. LDH Assay

To assess the protective effect of RSL against cell damage, an LDH assay (Abcam, Cambridge, UK) was conducted as per the guidelines provided by the manufacturer. Following treatment, cells were collected from the 96-well culture plates, and the assay reaction mixture was applied to each well. The optical density was analyzed at 450 nm using an Epoch microplate reader (Epoch), with readings taken every 2 min for 1 h.

### 2.6. Live–Dead Assay

To visually detect dead cells, a live and dead cell imaging kit (ThermoFisher Scientific, Waltham, MA, USA) was used following the manufacturer’s instructions. NP cells were stained with the color reagent included in the kit and incubated for 15 min at room temperature. Subsequently, the stained NP cells were imaged immediately using a confocal microscope (Nikon, Tokyo, Japan) at 100× magnification. The fluorescence intensity was analyzed using ImageJ software (v1.37, National Institutes of Health, Bethesda, MD, USA). The process began by minimizing non-specific stained fluorescence using the ‘Subtract Background’ function in ImageJ, which is crucial for accurately distinguishing positive and negative signals and quantifying live and dead cells. To measure the intensity of live cells (stained by green fluorescence) and dead cells (stained by red fluorescence), the ‘Mean Gray Value’ was used to calculate the average fluorescence intensity for each cell, providing a measure of live versus dead cells.

### 2.7. Western Blotting

Following the administration of treatments, cell homogenization was performed using RIPA buffer (GenDEPOT, Barker, TX, USA) supplemented with protease and phosphatase inhibitors (Millipore, Burlington, VT, USA). Protein levels in the lysates were determined using a Pierce™ BCA Protein Assay Kits (ThermoFisher Scientific, Waltham, MA, USA)) according to the instructions from the manufacturer. The proteins were electrophoresed on a 10–12% sodium dodecyl sulfate–polyacrylamide gel and transferred onto polyvinylidene difluoride membranes (Millipore). The membranes were blocked using 5% Difco™ Skim Milk (BD Biosciences, Franklin Lakes, NJ, USA) in Tris-buffered saline (Bio-Rad, Hercules, CA, USA) containing Tween 20 and incubated at 4 °C overnight with primary antibodies (Table 1). After membrane washing, it was exposed to secondary antibodies (Table 1) for 2 h at room temperature. Detection of protein bands was accomplished using an enhanced chemiluminescence system (Bio-Rad, Hercules, CA, USA) and results were visualized using an Amersham Imager 600 (GE Healthcare Life Sciences, Uppsala, Sweden).

### 2.8. Real-Time PCR

After treatment with RSL, the TRIzol reagent (ThermoFisher Scientific, Waltham, MA, USA) was added to extract cellular RNA from each culture well. Extracted RNA was used to synthesize complementary DNA (cDNA) using Accupower RT PreMix and oligo-dT primers (Bioneer, Daejeon, Republic of Korea). Quantitative real-time PCR analyses were conducted using SYBR Green Master Mix (Bioneer, Daejeon, Republic of Korea), cDNA, and primers in a CFX Connect Real-Time PCR Detection system (Bio-Rad, Hercules, CA, USA). The primer sequences used in this research are provided in Table 2.

### 2.9. Apoptosis

To quantify the percentage of apoptotic cells, we used the Annexin V Apoptosis Detection Kit (BioLegend, San Diego, CA, USA) following the instructions detailed by the manufacturer. Briefly, cells were separated from the plate using trypsin/EDTA and washed. The cells were stained with Annexin V binding buffer, APC Annexin V solution, and propidium iodide solution at room temperature away in the dark. Data were analyzed with an Accuri™ C6 Plus Flow Cytometer (BD Biosciences, Franklin Lakes, NJ, USA), and data processing was carried out using FlowJo software v10 (BD Biosciences, Franklin Lakes, NJ, USA).

### 2.10. Immunocytochemistry

Following treatment, cells underwent fixation with 4% paraformaldehyde at room temperature and were subsequently permeabilized with 0.2% Triton X-100 in PBS. After washing twice, the cells were incubated in 2% (*v*/*v*) normal goat serum for 1 h to block non-specific binding. The cells were then incubated with the primary antibodies listed in Table 1 at 4 °C overnight. Immediately after incubation, cells were stimulated with a fluorescent secondary antibody (Jackson ImmunoResearch, West Grove, PA, USA) for 2 h at room temperature. Finally, the nuclei were stained with 1 μg/mL of 4′, 6-diamidino-2-phenylindole (DAPI, Tokyo Chemical Industry, Japan). Imaging of these cells was performed using a confocal microscope (Nikon, Tokyo, Japan) at 400× magnification with settings consistently applied across all experiments (primary antibodies: laser power = 4, gain power = 7/DAPI: laser power = 5, gain power = 7), and fluorescence intensity was measured utilizing ImageJ software v1.54 (NIH, Frederick, MD, USA). To measure the fluorescence intensity of ACAN, Col2a1, MMP3, ADAMTS-5, and TREM2, quantitative analysis was performed using ImageJ software. Non-specific stained fluorescence was minimized using the ‘Subtract Background’ function in ImageJ. The ‘Mean Gray Value’ option was selected in the ‘Set Measurements’ menu, and the fluorescence intensity was measured accordingly. The intensity values were then calculated using the ‘Mean Gray Value’.

### 2.11. Statistical Analyses

Data were statistically analyzed and are presented as mean ± SEM using the GraphPad Prism software v8 (California, Boston, MA, USA). To compare between groups, a one-way ANOVA was utilized, with subsequent Tukey’s post hoc test applied to pinpoint significant differences.

## 3. Results

### 3.1. RSL Extracts Protect Human NP Cells against H_2_O_2_ Treatment

We evaluated whether the RSL extracts could prevent the compromised cell viability and NP cell damage induced by H_2_O_2_ depending on the concentration used. We used the CCK8, lactate dehydrogenase *(*LDH*)* assay, and live and dead imaging. One day after culturing, NP cells received treatments with various concentrations of RSL without H*_2_*O*_2_* addition. Subsequent measurements were recorded 24 h after treatments. RSL, up to 400 µg/mL, was nontoxic to NP cells and the cell viability was significantly increased starting from 50 µg/mL (Figure 1A). The cell viability revealed significantly compromised cells after H_2_O_2_ treatment compared to untreated cells. However, RSL treatment notably improved cell viability when used at 25–400 µg/mL (Figure 1B). We selected the three concentrations of the RSL extract—25, 100, and 400 µg/mL—and evaluated their protective effects against H_2_O_2_ challenge by assaying for lactate dehydrogenase (LDH) in NP cell culture media. LDH is released when cells are damaged. LDH activity was significantly increased by H_2_O_2_ treatment, whereas LDH activity was notably reduced in NP cells following treatment with 100 or 400 µg/mL of the RSL extracts (Figure 1C). Additionally, improved cell viability observed by the CCK8 analysis was further confirmed using live–dead imaging. Live–dead imaging showed that exposure to H_2_O_2_ increased the number of red-stained, dead cells. However, cell death was significantly reduced with RSL treatment at 100 and 400 µg/mL against H_2_O_2_ (Figure 1D,E).

### 3.2. RSL Extracts Inhibit Apoptosis in H_2_O_2_-Treated NP Cells

To investigate the pharmacological mechanisms behind RSL’s protective effects on oxidative-stressed NP cells, we measured the protein levels of cleaved caspase-3, Bax, and Bcl-2 by Western blot (Figure 2A). Bax promotes apoptosis by release of cytochrome c and caspase-3 activation [31,32]. Cleaved caspase-3 and Bax expression significantly increased in the H_2_O_2_ group compared to the blank group. In contrast, cells treated with RSL displayed dose-dependent decreases in these protein levels. Notably, a significant difference was observed in cells treated with RSL (100 and 400 µg/mL) under H_2_O_2_ conditions (Figure 2B,C). The level of Bcl-2, an anti-apoptotic protein, was significantly decreased after treatment with H_2_O_2_. However, its levels tended to increase in a concentration-dependent manner following RSL treatment. However, a notable increase in Bcl-2 protein expression was observed only at the 400 µg/mL concentration of RSL (Figure 2D). Additionally, cell death caused by H_2_O_2_ treatment could be characterized as apoptosis or necrosis as analyzed by flow cytometry (Figure 2E). After H_2_O_2_ treatment, the number of Annexin V^+^ cells significantly increased, indicating strong induction of apoptosis. However, RSL treatment reduced apoptosis in a concentration-dependent manner, with notable effects observed at 100 and 400 µg/mL of RSL extracts compared to the H_2_O_2_ group (Figure 2F).

### 3.3. RSL Inhibits H_2_O_2_-Induced Degeneration in NP Cells by Regulating the Expression of Key Matrix Components and Enzyme-Related mRNAs

To determine if RSL extracts prevent degeneration in NP cells caused by H_2_O_2_, we analyzed the RNA levels of key markers, including aggrecan (ACAN), collagen type II alpha-1 chain (Col2a1), a disintegrin and metalloproteinase with thrombospondin motifs (ADAMTS), and matrix metalloproteinases (MMPs), which are critical components of the ECM in NP cells and represent cellular degeneration [33,34]. In H_2_O_2_-exposed NP cells, the expression levels of *ACAN* and *Col2a1* were significantly reduced. Conversely, treatment with 400 µg/mL of RSL markedly increased the expression levels of *ACAN* and *Col2a1*, suggesting a protective effect against H_2_O_2_-induced degeneration (Figure 3A,B). Further analysis revealed that the expression of catabolic enzymes such as *ADAMTS4*, *ADAMTS5*, *MMP3*, and *MMP13*, which contribute to matrix degradation, was significantly upregulated in response to H_2_O_2_ treatment. However, these increases were effectively mitigated by RSL treatment, suggesting that RSL extracts can inhibit the upregulation of these degenerative enzymes (Figure 3C–F). These results indicate that, except for ACAN, the other markers are adjusted to levels close to those of the blank group by the RSL400 treatment. This suggests that RSL extract can effectively regulate the expression of genes related to H_2_O_2_-induced degeneration of NP cells.

### 3.4. RSL Inhibits H_2_O_2_-Induced Degeneration in NP Cells by Regulating the Expression of Key Matrix Components and Enzyme-Related Proteins

To substantiate these findings at the protein level, we conducted immunocytochemistry (ICC) analyses of ACAN, Col2a1, MMP3, and ADAMTS-5. The ICC results confirm that the mRNA levels correlate with the corresponding protein levels (Figure 4A). Specifically, the expression intensity of ACAN was significantly reduced in the H_2_O_2_ group, while it showed a concentration-dependent and significant increase in the RSL groups (Figure 4B). For Col2a1, the expression level was significantly decreased after H_2_O_2_ treatment. However, upon RSL treatment, there was a concentration-dependent increase in Col2a1 expression, with significant differences observed at 100 and 400 µg/mL concentrations compared to the H_2_O_2_ group (Figure 4C). In contrast, the expression intensity of MMP3 was significantly elevated following H_2_O_2_ treatment. RSL treatment significantly reduced the expression levels of MMP3, demonstrating its inhibitory effect on this catabolic enzyme (Figure 4D). Similarly, ADAMTS-5 levels were markedly increased with H_2_O_2_ treatment, but RSL treatment led to a concentration-dependent and significant decrease in ADAMTS-5 expression (Figure 4E). These results collectively indicate that RSL extracts effectively regulate the expression of key extracellular matrix components and degenerative enzymes in H_2_O_2_-treated NP cells, supporting the protective role of RSL against cellular degeneration.

### 3.5. RSL Regulates TREM2 Expression in H_2_O_2_-Treated NP Cells

To investigate the regulatory effects of RSL extracts on TREM2 expression in human NP cells exposed to H_2_O_2_, we conducted a series of experiments measuring TREM2 levels at both the mRNA and protein levels. Initially, we employed ICC to assess TREM2 expression in these cells (Figure 5A). The ICC results demonstrated a significant increase in TREM2 protein localization and intensity in cells treated with H_2_O_2_ compared to the blank group, characterized by a punctate intracellular expression pattern. In contrast, TREM2 expression was notably reduced following treatment with RSL. Quantification of TREM2 intensity revealed that its expression was significantly elevated in the H_2_O_2_ group, whereas cells treated with RSL exhibited a dose-dependent and significant reduction in TREM2 expression (Figure 5B). Furthermore, changes in TREM2 expression were corroborated at the protein level via Western blot analysis. Consistent with the ICC findings, TREM2 expression significantly increased following H_2_O_2_ treatment and exhibited a significant concentration-dependent decrease after exposure to RSL (Figure 5C,D). We confirmed TREM2 expression at the mRNA level using real-time PCR, which revealed more pronounced differences between the groups. Specifically, TREM2 mRNA expression significantly increased in control cells treated with H_2_O_2_, whereas it significantly decreased in a concentration-dependent manner in cells with RSL treatment (Figure 5E). These findings suggest that RSL extracts have a regulatory effect on TREM2 expression in cells under oxidative stress conditions.

## 4. Discussion

NPs acts as a shock-absorbing cushions between the vertebral discs and are crucial for maintaining spinal integrity and function [8,35]. IDD, marked by a progressive reduction in NP cells, leads to diminishing intervertebral spacing and causes inflammation and pain [36]. Although the pharmacological effects of various drugs on NP cells have been extensively studied, nutraceuticals have scarcely been scrutinized. To identify herbal extracts that exert cytoprotective effects on human NP cells challenged by H_2_O_2_, we initially screened several candidates, including RSL, *Luffa cylindrica* Roem., *Lycium chinense* Mill., and *Euphoria longana* Lam. We found that the RSL extract particularly effectively enhanced the rate of surviving human NP cells against H_2_O_2_-induced oxidative stress. Initial CCK8 assays showed that RSL extracts alone significantly enhanced the human NP cells’ viability in a dose-dependent manner, beginning at a concentration of 50 µg/mL. Importantly, RSL was nontoxic up to 400 µg/mL, suggesting a safe dosage range for potential therapeutic uses against cytotoxicity induced by oxidative stress. Moreover, RSL can protect human NP cells against oxidative injury, as indicated by the significantly reduced cell viability, LDH activity, and live-to-dead cell ratios following H_2_O_2_ treatment compared to the untreated cells, along with the enhanced viability observed with the three optimal RSL concentrations (25, 100, and 400 µg/mL). Although our findings strongly suggest a protective effect of RSL, there is insufficient evidence to show that RSL directly scavenges H_2_O_2_. Therefore, further research is necessary to definitively determine the mechanism by which RSL reduces oxidative stress in NP cells.

In a follow-up study, we determined the anti-apoptotic effects of RSL on apoptosis to further detail the molecular mechanisms underlying its cytoprotective effects. The intrinsic apoptotic mechanisms reportedly influence NP cell viability and longevity, primarily by altering Bax, Bcl-2, caspase-3, collagen, and ACAN levels [37,38]. Bax, a pro-apoptotic protein, is activated by intracellular stresses, such as hypoxia, oxidative stress, or excessive mechanical loading, leading to its mitochondrial translocation [39]. Activated Bax triggers caspase-3 cleavage, initiating the execution phase of apoptosis, which involves breakdown of key proteins and cellular structures, ultimately causing cell death. Elevated levels of cleaved caspase-3, which correspond to high apoptosis rates, have been observed in degenerated tissues of discs [40,41]. Conversely, Bcl-2 counteracts the pro-apoptotic effects of Bax, modulating apoptosis of the NP cells [42,43]. In healthy discs, Bcl-2 prevents apoptosis, preserving the cellular integrity, thereby ensuring that cells survive under unfavorable conditions. However, in IDD, Bcl-2 expression is often reduced, diminishing its protective role and promoting apoptosis. Low Bcl-2 expression under degenerative conditions leads to increased apoptosis, contributing to loss of the viable NP cells [44,45,46]. Thus, a balance between pro-apoptotic and anti-apoptotic proteins is crucial for regulating the life cycle of NP cells. We have shown that RSL extracts significantly protect NP cells from H_2_O_2_-induced apoptosis though modulating the expression of the key pro-apoptotic proteins, such as caspase-3 and Bax, and anti-apoptotic proteins, such as Bcl-2. The reduction in apoptotic rates and altered expression of apoptosis-associated proteins highlight the potential of RSL as a cytoprotective agent in conditions characterized by increased apoptosis due to oxidative stress. 

However, even after using 400 µg/mL of RSL, cell viability did not return to 100%, as observed in the blank group (without H_2_O_2_ and RSL), suggesting that additional mechanisms could be required to fully restore cell viability. This indicates that while RSL reduces the levels of cleaved caspase-3, Bax, and Bcl-2 and decreases the percentage of apoptotic cells to near-normal values, it does not completely counteract the effects of the strong oxidative stress induced by H_2_O_2_. Therefore, RSL alone may not be sufficient to fully protect cells from oxidative stress. Combining RSL with other treatments could potentially offer better protection and help in completely mitigating the effects of H_2_O_2_, addressing mechanisms not targeted by RSL alone.

Additionally, we investigated the effects of RSL on the degeneration of NP cells at the mRNA level, especially regarding factors that maintain ECM, such as *ACAN*, *Col2a1*, *ADAMTS*, and *MMPs*. The NP tissue’s ECM mainly consists of collagen (types II, IX, and XI) and proteoglycans such as ACAN that essentially allow the NP substance to resist compression while maintaining hydration [8,47]. As IDD progresses, the synthesis of these crucial molecules is typically downregulated. Concurrently, ECM-degrading enzymes such as MMPs and ADAMTS are upregulated. These enzymes break down collagen and ACAN, thereby exacerbating ECM degradation [34]. We found that the RSL extracts can increase the expression of *ACAN* and *Col2a1* simultaneously, suppressing the expression of *MMPs* and *ADAMTS*. Thus, promoting anabolic activity and inhibiting catabolic enzymes can essentially reduce ECM degradation, potentially alleviating the effects of IDD. 

The TREM2 receptor exerts critical roles in human NP cells [27]. TREM2 downregulation may prevent NP cell degeneration by reducing apoptosis, enhancing cell proliferation, and minimizing inflammation. Conversely, excessive upregulation of TREM2 in normal NP cells may contribute to their degeneration. TREM2 acts by regulating the expression and nuclear translocation of NF-κB p65. Our findings indicate that RSL effectively reduces the protein and mRNA levels of TREM2, both of which were upregulated in response to H_2_O_2_, and that inhibiting TREM2 expression prevents apoptosis and further degeneration of NP cells. 

However, our study demonstrated the protective effects of RSL on NP cells, but one of the major limitations is that we did not investigate the effects on other cellular pathways that might potentially crosstalk with TREM2. Such crosstalk could influence the overall outcomes of RSL treatment and contribute to the observed effects on cell viability and apoptosis. Therefore, future research should explore the specific signaling pathways that may interact with TREM2 during RSL treatment in more detail. Additionally, our study lacks an evaluation of the long-term effects of RSL treatment and its efficacy when combined with other therapeutic agents. Future studies should address these aspects to maximize the clinical applicability of RSL.

## 5. Conclusions

This study highlights the protective effects of RSL on NP cells in the treatment of IDD. RSL protects NP cells from oxidative stress by modulating apoptosis-related proteins, increases the expression of ECM components, and inhibits ECM-degrading enzymes, thereby supporting disc integrity. Specifically, RSL’s effect on regulating the TREM2 signaling pathway activated under oxidative stress conditions was confirmed. These findings suggest that while RSL alone did not fully restore cell viability to blank levels, combining RSL with other therapies could provide better protection. Future research should investigate the additional mechanisms of RSL that may interact with TREM2, as well as its long-term effects and efficacy in combination with other treatments.

## Figures and Tables

**Figure 1 biology-13-00602-f001:**
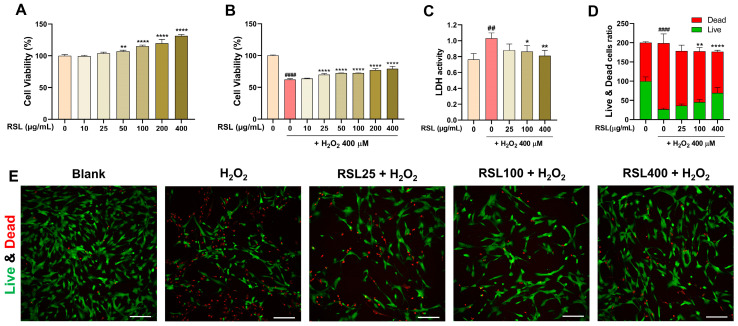
Cytoprotective effect of RSL against H_2_O_2_-induced oxidative stress of NP cells. (**A**) Cell viability of NP cells treated with RSL for 24 h, in the absence of H_2_O_2_ (400 µM), was demonstrated by CCK8 assay (*n* = 6). (**B**) Viability of NP cells, pretreated with RSL for 30 min followed by H_2_O_2_ treatment and then cultured for 24 h, was revealed by same method (*n* = 6). (**C**) LDH activity in culture media was derived from NP cells in each group (*n* = 6). (**D**) The intensity ratio of live or dead NP cells treated with RSL for 30 min before H_2_O_2_ treatment and then cultured for 24 h (*n* = 6). (**E**) The fluorescence micrographs of live–dead assay for live (green) or dead (red) cells in blank, H_2_O_2_, and RSL groups. White scale bar represents 200 μm. Data are expressed as mean ± standard error of the mean (SEM). Significant differences were determined using one-way analysis of variance (ANOVA) with Tukey’s post hoc test as follows: ^##^ *p* < 0.01 and ^####^ *p* < 0.0001 vs. the blank group; * *p* < 0.05, ** *p* < 0.01, and **** *p* < 0.0001 vs. the H_2_O_2_ group.

**Figure 2 biology-13-00602-f002:**
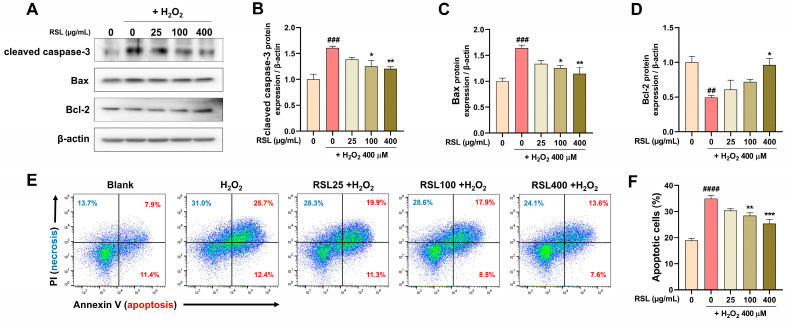
The effect of RSL on proteins related to apoptosis in H_2_O_2_-treated NP cells. (**A**) Protein expression of cleaved caspase-3, Bax, and Bcl-2 in Western blot analysis. (**B**–**D**) Quantitative representation of relative protein levels of cleaved caspase-3, Bax, and Bcl-2 (*n* = 4). (**E**) Flow-cytometric analysis of Annexin V-APC (red)/PI-PE (blue) double staining depicting cell death modalities in H_2_O_2_ (400 µM)-treated NP cells. (**F**) The percentages of Annexin V-positive cells in flow cytometry results (*n* = 4). Data are expressed as mean ± SEM. Statistical significance was assessed using one-way ANOVA with Tukey’s post hoc analysis as follows: ^##^ *p* < 0.01, ^###^ *p* < 0.001, and ^####^ *p* < 0.0001 vs. the blank group; * *p* < 0.05, ** *p* < 0.01, and *** *p* < 0.001 vs. the H_2_O_2_ group.

**Figure 3 biology-13-00602-f003:**
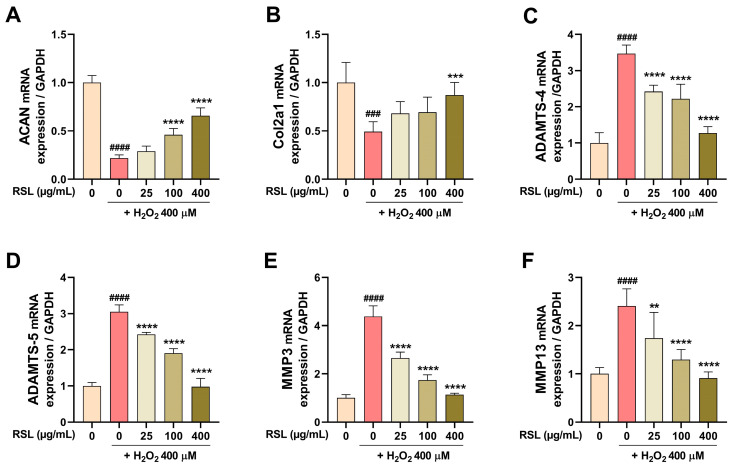
Effects of RSL on ECM component and degenerative enzyme expression in H_2_O_2_-treated NP Cells. Quantitative real-time PCR analysis of degeneration-associated genes in H_2_O_2_ (400 µM)-treated NP cells. (**A**) *ACAN*, (**B**) *Col2a1*, (**C**) *ADAMTS-4*, (**D**) *ADAMTS-5*, (**E**) *MMP3*, and (**F**) *MMP13* (*n* = 6). Statistical significance was assessed through one-way ANOVA followed by Tukey’s post hoc test as follows: ^###^ *p* < 0.001 and ^####^ *p* < 0.0001 vs. the blank group; ** *p* < 0.01, *** *p* < 0.001, and **** *p* < 0.0001 vs. the H_2_O_2_ group.

**Figure 4 biology-13-00602-f004:**
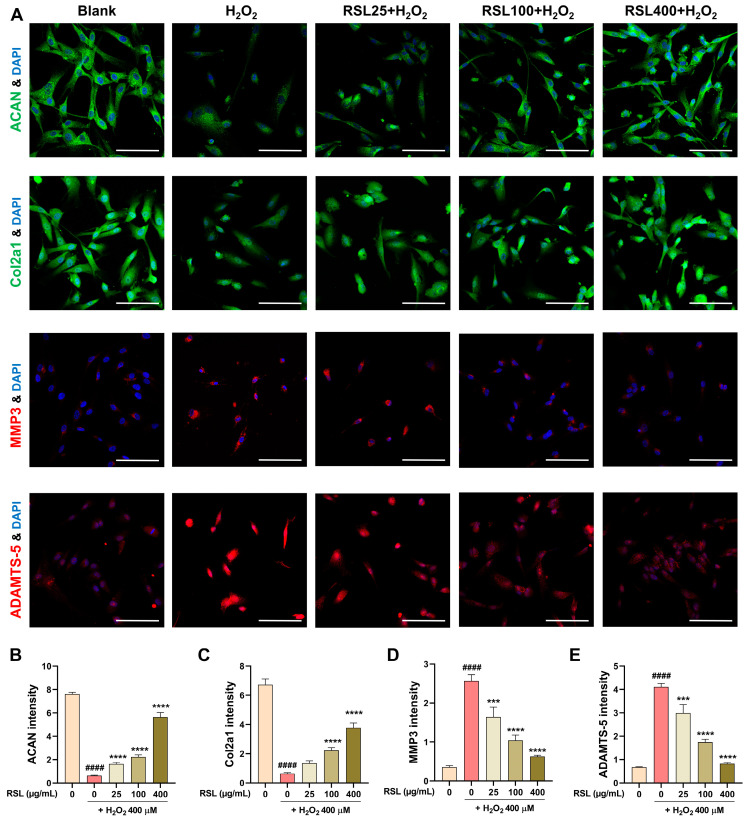
Immunocytochemistry analysis of ACAN, Col2a1, MMP3, and ADAMTS-5 protein levels in H_2_O_2_-treated NP cells with RSL treatment. (**A**) Representative immunocytochemistry (ICC) images showing ACAN (green), Col2a1 (green), MMP3 (red), and ADAMTS-5 (red) in NP cells. White scale bar represents 100 μm. (**B**–**E**) Quantification of ICC analysis of ACAN, Col2a1, MMP3, and ADAMTS-5 expression intensity in H_2_O_2_ (400 µM)-treated NP cells with RSL treatment. Statistical significance was assessed using one-way ANOVA with Tukey’s post hoc analysis as follows: ^####^ *p* < 0.0001 vs. the blank group; *** *p* < 0.001, and **** *p* < 0.0001 vs. the H_2_O_2_ group.

**Figure 5 biology-13-00602-f005:**
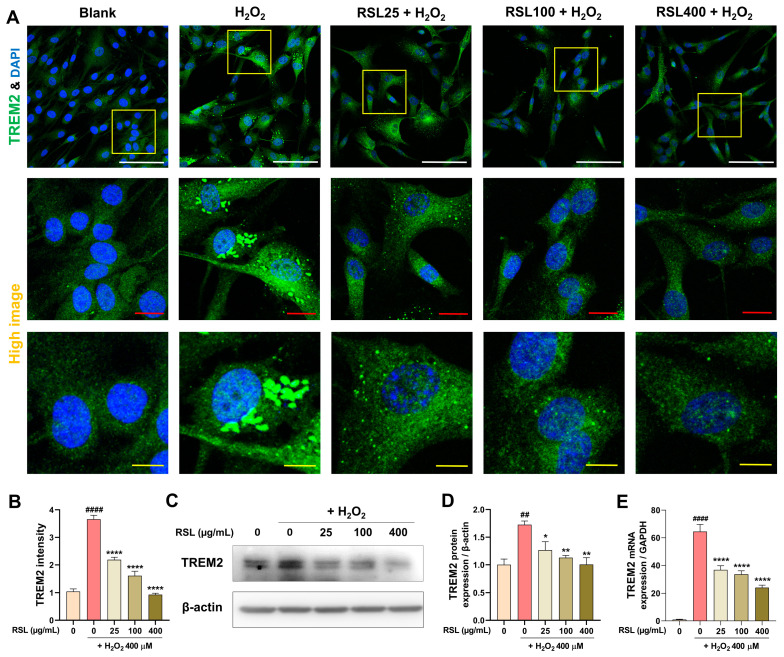
The regulation of TREM2 expression by RSL in H_2_O_2_-treated NP cells. (**A,B**) Immunocytochemistry images and quantification of relative intensity showing expression of TREM2 protein in H_2_O_2_-treated NP cells. The yellow box represents an enlarged section to show the distribution of TREM2 within the cell. White scale bar represents 100 μm, red scale bar represents 10 μm, and yellow scale bar represents 5 μm. (**C,D**) Western blot analysis confirming changes in TREM2 protein levels. (**E**) mRNA levels of TREM2 expression using a real-time PCR. Data are expressed as mean ± SEM. Statistical significance was assessed using one-way ANOVA with Tukey’s post hoc analysis as follows: ^##^ *p* < 0.01 and ^####^ *p* < 0.0001 vs. the blank group; * *p* < 0.05, ** *p* < 0.01, and **** *p* < 0.0001 vs. the H_2_O_2_ group.

**Table 1 biology-13-00602-t001:** Primary antibodies used in Western blotting and immunocytochemistry.

	Company	Product No.	Host	Reactivity	Application	Dilution
Cleaved caspase-3	CST	9661	Rabbit	H M R Mk	WB	1:500
Bax	CST	2772	Rabbit	H M R Mk	WB	1:1000
BCL-2	Santa cruz	SC-7382	Mouse	H M R	WB	1:200
TREM2	Santa cruz	SC-373828	Mouse	H	WB	1:200
β-actin	Santa cruz	SC-47778	Mouse	H M R	WB	1:3000
Aggrecan	Proteintech	13880-1-AP	Rabbit	H M R	ICC	1:200
Col2a1	Invtirogen	PA1-26206	Rabbit	B H R	ICC	1:100
Adamts5	Invtirogen	PA5-27165	Rabbit	H M	ICC	1:100
MMP3	Abcam	Ab52915	Rabbit	H M R	ICC	1:100
FITC Goat anti-rab IgG	Jackson	112-095-003	Goat		ICC	1:300
Rhodamine Goat anti-rab IgG	Jackson	111-295-045	Goat		ICC	1:300
Goat anti-rabbit IgG	Abcam	ab205718	Goat		WB	1:2500
Goat anti-mouse IgG	Abcam	ab205719	Goat		WB	1:2500

**Table 2 biology-13-00602-t002:** The sequences of the primers used in real-time PCR.

Target	Forward (5′–3′)	Reverse (3′–5′)
*ACAN*	TGAAACCACCTCTGCATTCCA	GACGCCTCGCCTTCTTGAA
*Col2a1*	GTCACAGAAGACCTCACGCCTC	TCCACACCGAATTCCTGCTC
*Adamts* *-4*	ACTGGTGGTGGCAGATGACA	TCACTGTTAGCAGGTAGCGCTTT
*Adamts* *-5*	GCTTCTATCGGGGCACAGT	CAGCAGTGGCTTTAGGGTGTAG
*MMP3*	GCTGTTTTTGAAGAATTTGGGTTC	GCACAGGCAGGAGAAAACGA
*MMP13*	CCAGGCATCACCATTCAAG	ATCATCTTCATCACCACCACTG

## Data Availability

The data presented in this study are available upon request from the corresponding author.

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
