# Peer review of "Raphanus sativus Linne Protects Human Nucleus Pulposus Cells against H2O2-Induced Damage by Inhibiting TREM2"

_biology, 2024, doi:10.3390/biology13080602_

Round 1

Reviewer 1 Report

Comments and Suggestions for Authors

The manuscript by Kim et al. describes a study the Raphanus Sativus Linne protects the human nucleus pulposus cells against H2O2-induced damage. The authors demonstrate that RSL enhances the viability of NP cells and mitigates their degeneration by modulating the expression of the TERM2 gene. The authors are commended for an interesting study. The manuscript is well organized and conclusive. Specific comments/suggestions to further improve the manuscript prior to acceptance are as follows:

1.    In this study, the authors exposed the cells to H2O2 for a duration of 24 hours. It raises the question: what would be the outcome if the cells were subjected to a shorter treatment period, for instance, 30 minutes?

2.    In figure 1, panel E, how was the data analyzed? And how did the authors distinguish and count live cells when they are in close proximity to each other? Please include H2O2 in the labeling along with the varying concentrations of RSL for clarification.

3.    Page 6, lines 228 and 232. Page 7, lines 251-255, add references.

4.    Figure 2. Just by looking at the WB image in panel, it appears that the group treated with 100 ug/ml RSL exhibits a higher level of cleaved caspase-3 compared to the 25 ug/ml RSL group. Could the authors please clarify?

5.    Page 3, section 3.3. The authors examined the gene expression of the genes associated with NP cell degeneration at mRNA level, however, the mRNA may not be translated into protein function. They should also examine the gene expression at protein level.

6.    In figure 4, panel A, add scale bar to the zoomed-in images.

7.    In the methods section, please provide a detailed description of how the intensity of TERM was measured and analyzed?

8.    In the discussion, the authors acknowledged a limitation of the study: the absence of an in-depth examination of the specific pathways involved. It would be beneficial if the authors could expand on this topic. For instance, they could explore and discuss potential pathways that might play a crucial role in this process.

Comments on the Quality of English Language

The English is satisfactory, requiring only minor edits.

Author Response

We wish to resubmit the manuscript titled “Raphanus Sativus Linne Protects the Human Nucleus Pulposus Cells against H2O2-Induced Damage by Inhibiting TREM2”

We thank the editor and reviewers for their excellent and constructive comments, which clearly helped to improve the quality of this manuscript. We are pleased to provide the following point-by-point reply. Appropriate changes suggested by the reviewers have been introduced into the manuscript (highlighted within the manuscript).

We hope that our manuscript will be acceptable for publication in Biology

Kind regards,

In-Hyuk Ha, M.D., Ph.D.

Reviewer 2 Report

Comments and Suggestions for Authors

Raphanus Sativus Linne Protects the Human Nucleus pulposus cells against H2O2-indcued damage by inhibiting TREM2 showed that RSL in high concentration protects cell culture from the programmed death triggered by H2O2. Nucleus pulposus is the core of the intervertebral disc. The degeneration of nucleus pulposus is considered the key part to maintain the health of the intervertebral disc. Therefore, the method to improve the nucleus pulposus health is desired. The result in this paper is interesting in this scenario. Through multiple methods, the authors showed that increased concentration of RSL dramatically increased the viability of the cell cultured w/ hydrogen peroxide added. The paper further showed that the apoptosis decreased in these cultured cells w/ RSL added. The apoptotic cascade was blocked from the beginning.

ECM in cultured NP cells is protected by RSL administration. Is it possible that RSL directly protects the ECM oxidation? Is there any evidence to show whether RSL protects ECM or directly eliminates H2O2? Is there any test which shows the concentration of active oxygen after the addition of RSL?

Interestingly, the authors found that TREM2 in NP cells is downregulated. But it is unclear whether TREM2 is involved in NP cell apoptosis. If the authors can show its apoptotic function, it will be more meaningful.

Comments on the Quality of English Language

English is OK.

Author Response

(The authors gave the same response as above.)

Reviewer 3 Report

Comments and Suggestions for Authors

Manuscript title: "Raphanus Sativus Linne Protects the Human Nucleus Pulposus Cells against H2O2-Induced Damage by Inhibiting TREM2"

General comments:

The manuscript presents a study on the protective effects of Raphanus Sativus Linne (RSL) extract against oxidative stress in human nucleus pulposus (NP) cells. The study has implications in the Intervertebral Disc Degeneration (IDD), caused due to damage and depletion of NP cells. The manuscript is well written and provides a clear connection between the cyto-protective effects of RSL extract and its inhibitory effect on apoptosis. The work has been done in the context of NP cells, where authors inhibited TREM2, a pro-apoptotic protein.

However, there are some major corrections where the manuscript could be improved to enhance clarity before publication. These include aspects related to experimental design, data presentation, and interpretation.

Specific comments:

Abstract:

line 21-22: In contrast**, RSL extract increased Bcl2 concentration to downregulate apoptosis would sound better than using contrastively.

Introduction:

The introduction sets the stage by outlining the key problem of how damage to NP cells due to oxidative stress and mechanical pressure can trigger IDD. However, the link between oxidative stress and TREM2 function could be further strengthened to show the gap in the knowledge.

It would be beneficial to include more background on how TREM2 functions in the context of IDD and why it is a significant target molecule for the study.

line 37: reference #8 seems to be not properly formatted in the bibliography?

Materials and Methods:

line 94: How does cell viability compare in late passage cells? Do late-passage cells typically experience more oxidative stress compared to early-passage cells?

line 110: The manuscript would benefit from the explanation of selecting 400 µM of H2O2. Similar to the dose-dependent study of RSL extract, it would be helpful to include data showing how different concentrations of H2O2 affect cell viability. If the optimal concentration of H2O2 has already been established in previous studies using similar or other cell types, please cite those studies to support the author’s choice.

line 152 (table 1): Primary and secondary**. I see goat α-rabbit and goat α-mouse which are secondary antibodies? Please be sure that all antibodies are validated for specificity and cross-reactivity.

line 155: Be consistent: Thermo Fisher Scientific, MA, USA.

line 180: What was the working concentration of DAPI ?

line 181: What was the laser intensity and exposure time? The author can write this as “not more than 10% and 50 msec”, if values are variable in different experiments otherwise be specific.

Results:

All figure legends from Fig 1-4 must have the concentration of H2O2.

Fig 1 legend: Does n represent no of cells or no of biological repeats?

Fig 1b: Was there a significant difference between blank and 400 µg/ml of RSL extract? 

Fig 1e: Please label micrographs as RSL (25µg/ml and so on) for easy readability.

line 217 Fig 1e legend: Fluorescence micrographs**

Fig 2a: RSL seems to be in a different font compared to Cleaved caspase 3, Bax and others. Please be consistent.

Fig 2b, c, d, and f: Was there significant differences between blank and 400 µg/ml RSL in each case?The author show that even after using 400 µg/ml of RSL, Caspase3, Bax, Blc2 and % apoptotic cells didn’t go back like the blank cells (without RSL and H2O2)? Does that mean additional mechanisms? Maybe RSL can't entirely counteract the effects of strong oxidative stress of H2O2? and might need to be combined with other treatments to fully protect cells? 

Fig 3a, b, c: Was there significant differences between blank and 400 µg/ml in each case?

line 264: inhibiting**

Fig 4 general: Can the authors validate the result in Fig 4 with an assay by using a specific inhibitor or blocker of TREM2? This will show if the observed effects on viability and apoptosis are indeed mediated through TREM2 pathways.

Fig 4a: Please label micrographs as RSL (25µg/ml and so on) for easy readability.

Fig 4a legend: State what is depicted in the white box?

Fig 4e: Is there a significant difference between blank and 400 µg/ml?

line 312: Reference #8 is not properly formatted in bibliography?

The micrographs in Fig 4a are small and may be difficult to interpret.

Discussion:

The author shows that even after using 400 µg/ml of RSL, cell viability didn’t return to 100% as the blank (without H2O2 and RSL), suggesting additional mechanisms could be required to restore cell viability. This can also be addressed through a textual correction by adding in the discussion. Line 344-346: “We have shown that…” can be revised, based on the significant differences between blank and 400 µg/ml of RSL. Because even after adding 400 µg/ml RSL, it didn’t return to levels as the blank, this could suggest mechanisms not contributed by RSL.

Please discuss potential limitations of the study more thoroughly. For example, the effects of RSL on other cellular pathways which might potentially crosstalk with TREM2.

It would also be beneficial to speculate on the relevance of the discovery. 

Conclusion:

Consider writing in few sentence about the potential implications of this findings for developing new therapeutics for IDD.

Author Response

(The authors gave the same response as above.)

Round 2

Reviewer 3 Report

Comments and Suggestions for Authors

Line 28-29: The sentence seems confusing to readers. Maybe using "We administered hydrogen peroxide (H2O2) to cultured human NP cells treated with RSL extracts"?

Line 214: 1:1000 dilution of how much stock conc. (1mg/ml)? Please be precise.

Line 278: in a concentration-dependent manner?

Author Response

Comments 1: Line 28-29: The sentence seems confusing to readers. Maybe using "We administered hydrogen peroxide (H2O2) to cultured human NP cells treated with RSL extracts"?’

Response) Thank you very much for your comments. The sentence was changed as advised.

Comments 2: Line 214: 1:1000 dilution of how much stock conc. (1mg/ml)? Please be precise.

Response) We apologize for not entering the information accurately. Concentration and company information have also been added.

“the nuclei were stained with 1 μg/mL of 4′, 6-diamidino-2-phenylindole (DAPI, Tokyo Chemical Industry, Japan)”

Comments 3: Line 278: in a concentration-dependent manner?

Response) The sentence was changed as advised. We sincerely appreciate all your valuable comments and effort to review the manuscript.